# DREAMTIME: AN IMPROVED OPTIMIZATION STRATEGY FOR DIFFUSION-GUIDED 3D GENERATION

**Yukun Huang**[1,2*†], **Jianan Wang**[1*‡], **Yukai Shi**[1], **Boshi Tang**[1], **Xianbiao Qi**[1], **Lei Zhang**[1]
[1]International Digital Economy Academy (IDEA)
[2]The University of Hong Kong
{huangyukun,wangjianan,shiyukai,boshitang,qixianbiao,leizhang}@idea.edu.cn

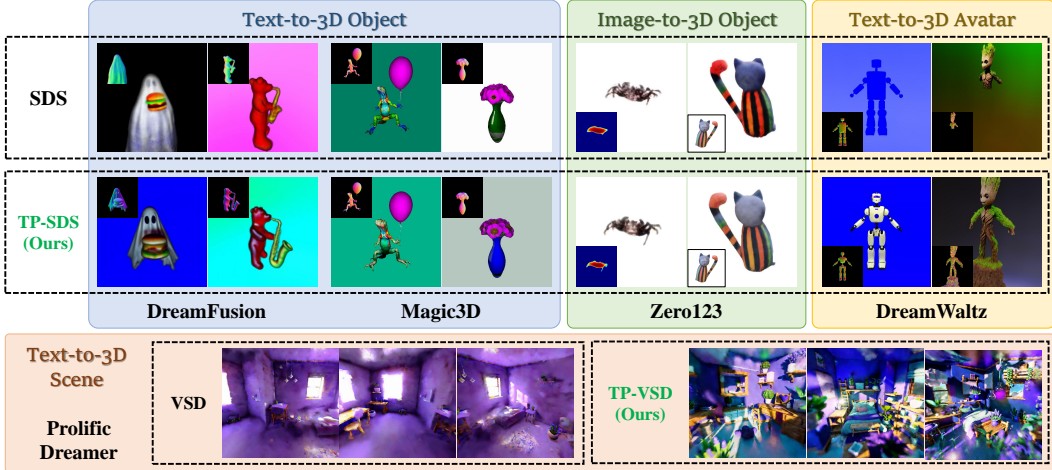

Figure 1: DreamTime enables higher-quality 3D generation across different methods, including: DreamFusion Poole et al. (2022), Magic3D Lin et al. (2022), Zero123 Liu et al. (2023a), DreamWaltz Huang et al. (2023), and ProlificDreamer Wang et al. (2023).

## ABSTRACT

Text-to-image diffusion models pre-trained on billions of image-text pairs have recently enabled 3D content creation by optimizing a randomly initialized differentiable 3D representation with score distillation. However, the optimization process suffers slow convergence and the resultant 3D models often exhibit two limitations: (a) quality concerns such as missing attributes and distorted shape and texture; (b) extremely low diversity comparing to text-guided image synthesis. In this paper, we show that the conflict between the 3D optimization process and uniform timestep sampling in score distillation is the main reason for these limitations. To resolve this conflict, we propose to prioritize timestep sampling with monotonically non-increasing functions, which aligns the 3D optimization process with the sampling process of diffusion model. Extensive experiments show that our simple redesign significantly improves 3D content creation with faster convergence, better quality and diversity.

## 1 INTRODUCTION

Humans are situated in a 3D environment. To simulate this experience for entertainment or research, we require a significant number of 3D assets to populate virtual environments like games and robotics simulations. Generating such 3D content is both expensive and time-consuming, necessitating skilled artists with extensive aesthetic and 3D modeling knowledge. It's reasonable to

---

*Equal contribution.
†Work done during an internship at IDEA.
‡Corresponding author.

inquire whether we can enhance this procedure to make it less arduous and allow beginners to create 3D content that reflects their own experiences and aesthetic preferences.

Recent advancements in text-to-image generation (Ramesh et al., 2022; Saharia et al., 2022; Rombach et al., 2022) have democratized image creation, enabled by large-scale image-text datasets, *e.g.*, Laion5B (Schuhmann et al., 2022), scraped from the internet. However, 3D data is not as easily accessible, making 3D generation with 2D supervision very attractive. Previous works (Poole et al., 2022; Wang et al., 2022; Lin et al., 2022; Metzer et al., 2022; Chen et al., 2023) have utilized pre-trained text-to-image diffusion models as a strong image prior to supervise 2D renderings of 3D models, with promising showcases for text-to-3D generation. However, the generated 3D models are often in low quality with unrealistic appearance, mainly due to the fact that text-to-image models are not able to produce identity-consistent object across multiple generations, neither can it provide camera pose-sensitive supervision required for optimizing a high-quality 3D representation. To mitigate such supervision conflicts, later works orthogonal to ours have explored using generative models with novel view synthesis capability (Liu et al., 2023a;b) or adapting pre-trained model to be aware of camera pose and current generation (Wang et al., 2023). However, challenges remain for creative 3D content creation, as the generation process still suffers from some combination of the following limitations as illustrated in Fig. 2: (1) requiring a long optimization time to generate a 3D object (slow convergence); (2) low generation quality such as missing text-implied attributes and distorted shape and texture; (3) low generation diversity.

As a class of score-based generative models (Ho et al., 2020; Song & Ermon, 2019; Song et al., 2021b), diffusion models contain a data noising and a data denoising process according to a predefined schedule over fixed number of timesteps. They model the denoising score $\nabla_{\mathbf{x}} \log p_{\text{data}}(\mathbf{x})$, which is the gradient of the log-density function with respect to the data on a large number of noise-perturbed data distributions. Each timestep ($t$) corresponds to a fixed noise with the score containing coarse-to-fine information as $t$ decreases. For image synthesis, the sampling process respects the discipline of coarse-to-fine content creation by iteratively refining samples with monotonically decreasing $t$. However, the recent works leveraging pre-trained data scores for diffusion-guided 3D generation (Poole et al., 2022; Wang et al., 2022; Lin et al., 2022; Chen et al., 2023; Qian et al., 2023) randomly sample $t$ during the process of 3D model optimization, which is counter-intuitive.

In this paper, we first investigate what a 3D model learns from pre-trained diffusion models at each noise level. Our key intuition is that pre-trained diffusion models provide different levels of visual concepts for different noise levels. At 3D model initialization, it needs coarse high-level information for structure formation. Later optimization steps should instead focus on refining details for better visual quality. These observations motivate us to propose time prioritized score distillation sampling (TP-SDS) for diffusion-guided 3D generation, which aims to prioritize information from different diffusion timesteps ($t$) at different stages of 3D optimization. More concretely, we propose a non-increasing timestep sampling strategy: at the beginning of optimization, we prioritize the sampling of large $t$ for guidance on global structure, and then gracefully decrease $t$ with training iteration to get more information on visual details. To validate the effectiveness of the proposed TP-SDS, we first analyze the score distillation process illustrated on 2D examples. We then evaluate TP-SDS against standard SDS on a wide range of text-to-3D and image-to-3D generations in comparison for convergence speed as well as model quality and diversity.

Our main contributions are as follows:

- We thoroughly reveal the conflict between diffusion-guided 3D optimization and uniform timestep sampling of score distillation sampling (SDS) from three perspectives: mathematical formulation, supervision misalignment, and out-of-distribution (OOD) score estimation.

- To resolve this conflict, we introduce DreamTime, an improved optimization strategy for diffusion-guided 3D content creation. Concretely, we propose to use a non-increasing time sampling strategy instead of uniform time sampling. The introduced strategy is simple but effective by aligning the 3D optimization process with the sampling process of DDPM (Ho et al., 2020).

- We conduct extensive experiments and show that our simple redesign of the optimization process significantly improves diffusion-guided 3D generation with faster convergence, better quality and diversity across different foundation diffusion models and 3D representations.

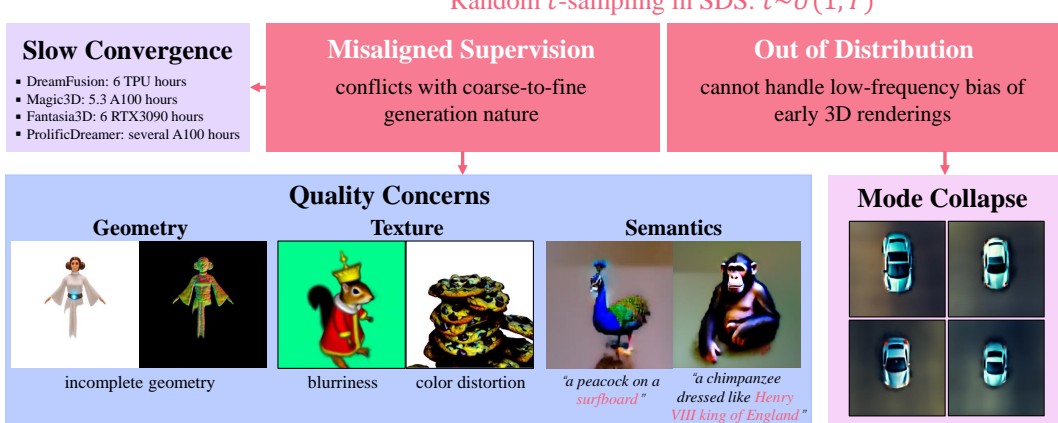

Figure 2: Challenging phenomena in optimization-based diffusion-guided 3D generation.

## 2 RELATED WORK

**Text-to-image generation.** Text-to-image models such as GLIDE (Nichol et al., 2021), un-CLIP (Ramesh et al., 2022), Imagen (Saharia et al., 2022), and Stable Diffusion (Rombach et al., 2022) have demonstrated impressive capability of generating photorealistic and creative images given textual instructions. The remarkable progress is enabled by advances in modeling such as diffusion models (Dhariwal & Nichol, 2021; Song et al., 2021a; Nichol & Dhariwal, 2021), as well as large-scale web data curation exceeding billions of image-text pairs (Schuhmann et al., 2022; Sharma et al., 2018; Changpinyo et al., 2021). Such datasets have wide coverage of general objects, likely containing instances with great variety such as color, texture and camera viewpoints. As a result, text-to-image diffusion models pre-trained on those billions of image-text pairs exhibit remarkable understanding of general objects, good enough to synthesize them with high quality and diversity. Recently, generating different viewpoints of the same object has made significant progresses, notably novel view synthesis from a single image (Liu et al., 2023a;b), which can be applied to improve the quality of image-to-3D generation, orthogonal to our approach.

**Diffusion-guided 3D generation.** The pioneering works of Dream Fields (Jain et al., 2022) and CLIPmesh (Mohammad Khalid et al., 2022) utilize CLIP (Radford et al., 2021) to optimize a 3D representation so that its 2D renderings align well with user-provided text prompt, without requiring expensive 3D training data. However, this approach tends to produce less realistic 3D models because CLIP only offers discriminative supervision on high-level semantics. In contrast, recent studies (Poole et al., 2022; Lin et al., 2022; Chen et al., 2023; Wang et al., 2023; Qian et al., 2023; Tang et al., 2023) have demonstrated remarkable text-to-3D and image-to-3D generation results by employing powerful pre-trained diffusion models as a robust 2D prior. We build upon this line of work and improve over the design choice of 3D model optimization process to enable significantly higher-fidelity and higher-diversity diffusion-guided 3D generation with faster convergence.

## 3 METHOD

We first review diffusion-guided 3D generation modules, including differentiable 3D representation (Mildenhall et al., 2021), diffusion models (Ho et al., 2020), and SDS (Poole et al., 2022) in Section 3.1. Then, we analyze the existing drawbacks of SDS in Section 3.2. Finally, to alleviate the problems in SDS, we introduce an improved optimization strategy in Section 3.3.

### 3.1 PRELIMINARY

**Differentiable 3D representation.** We aim at generating a 3D model $\theta$ that when rendered at any random view $c$, produces an image $\mathbf{x} = g(\theta, c)$ that is highly plausible as evaluated by a pre-trained text-to-image or image-to-image diffusion model. To be able to optimize such a 3D model, we require the 3D representation to be differentiable, such as NeRF (Mildenhall et al., 2021; Müller et al., 2022), NeuS (Wang et al., 2021) and DMTet (Shen et al., 2021).

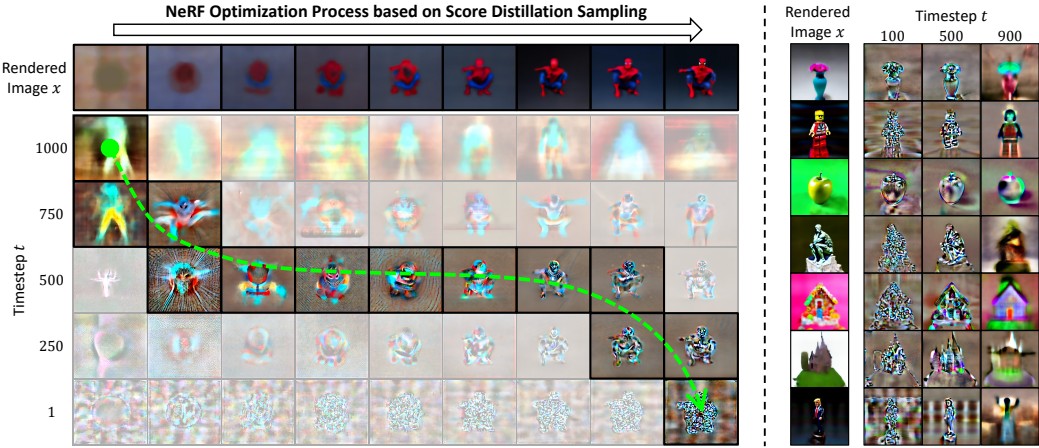

Figure 3: Visualization of SDS gradients under different timesteps $t$. (Left) Visualization of SDS gradients throughout the 3D model (in this case NeRF) optimization process, where the green curved arrow denotes the path for more informative gradient directions as NeRF optimization progresses. It can be observed that a non-increasing timestep $t$ is more suitable for the SDS optimization process. (Right) We provide more examples to illustrate the effects of timestep $t$ on SDS gradients. Small $t$ provides guidance on local details, while large $t$ is responsible for global structure.

**Diffusion models** (Ho et al., 2020; Nichol & Dhariwal, 2021) estimate the denoising score $\nabla_{\mathbf{x}} \log p_{\text{data}}(\mathbf{x})$ by adding noise to clean data $\mathbf{x} \sim p(\mathbf{x})$ in $T$ timesteps with pre-defined schedule $\alpha_t \in (0,1)$ and $\bar{\alpha}_t := \prod_{s=1}^{t} \alpha_s$, according to:

$$\mathbf{x}_t = \sqrt{\bar{\alpha}_t}\mathbf{x} + \sqrt{1-\bar{\alpha}_t}\boldsymbol{\epsilon}, \text{ where } \boldsymbol{\epsilon} \sim \mathcal{N}(\mathbf{0}, \mathbf{I}), \tag{1}$$

then learns to denoise by minimizing the noise prediction error. In the sampling stage, one can derive $\mathbf{x}$ from noisy input and noise prediction, and subsequently the score of data distribution.

**Score Distillation Sampling (SDS)** (Poole et al., 2022; Lin et al., 2022; Metzer et al., 2022) is a widely used method to distill 2D image priors from a pre-trained diffusion model $\boldsymbol{\epsilon}_{\phi}$ into a differentiable 3D representation. SDS calculates the gradients of the model parameters $\boldsymbol{\theta}$ by:

$$\nabla_{\boldsymbol{\theta}}\mathcal{L}_{\text{SDS}}(\phi, \mathbf{x}) = \mathbb{E}_{t,\boldsymbol{\epsilon}}\left[w(t)(\boldsymbol{\epsilon}_{\phi}(\mathbf{x}_t; y, t) - \boldsymbol{\epsilon})\frac{\partial \mathbf{x}}{\partial \boldsymbol{\theta}}\right], \tag{2}$$

where $w(t)$ is a weighting function that depends on the timestep $t$ and $y$ denotes a given text or image prompt. SDS optimization is robust to the choice of $w(t)$ as mentioned in Poole et al. (2022).

**Remark.** Our goal is to optimize a differentiable 3D representation by distilling knowledge from pre-trained Stable Diffusion (Rombach et al., 2022) or Zero123 (Liu et al., 2023a) given a text or image prompt, respectively. In the training process, SDS is used to supervise the distillation process.

## 3.2 ANALYSIS OF EXISTING DRAWBACKS IN SDS

A diffusion model generates an image by sequentially denoising a noisy image, where the denoising signal provides different granularity of information at different timestep $t$, from structure to details (Choi et al., 2022; Balaji et al., 2022). For diffusion-guided 3D content generation, however, SDS (Poole et al., 2022) samples $t$ from a uniform distribution throughout the 3D model optimization process, which is counter-intuitive because the nature of 3D generation is closer to DDPM sampling (sequential $t$-sampling) than DDPM training (uniform $t$-sampling). This motivates us to explore the potential impact of uniform $t$-sampling on diffusion-guided 3D generation.

In this subsection, we analyze the drawbacks of SDS from three perspectives: mathematical formulation, supervision alignment, and out-of-distribution (OOD) score estimation.

**Mathematical formulation.** We contrast SDS loss:

$$\mathcal{L}_{\text{SDS}}(\phi, \mathbf{x}_t) = \mathbb{E}_{t \sim \mathcal{U}(1,T)}\left[w(t)\|\boldsymbol{\epsilon}_{\phi}(\mathbf{x}_t; y, t) - \boldsymbol{\epsilon}\|_2^2\right] \tag{3}$$

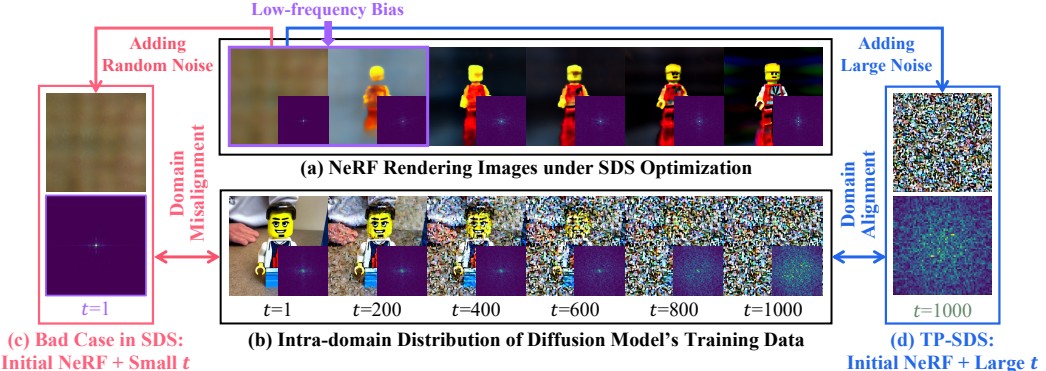

Figure 4: Illustration of OOD issue using web-data pre-trained diffusion model for denoising NeRF rendered images, in pixel and frequency domain. We provide an extreme case to show the frequency domain misalignment: (c) adding small noise to NeRF's rendering at initialization. (d) illustrates that TP-SDS avoids such domain gap by choosing the right noise level.

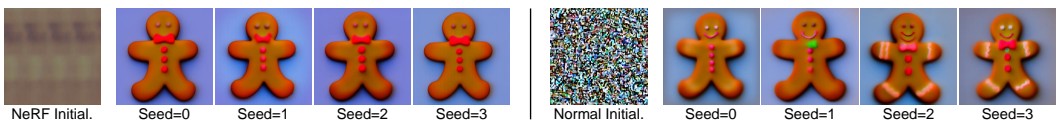

Figure 5: Low-frequency bias of initial rendered image leads to low-diversity generation. To demonstrate this, we provide SDS-optimized 2D generated results using text prompt "gingerbread man". Compared to normal initialization (Right), NeRF initialization (Left) exhibits low frequency and leads to the OOD issue, failing to produce diverse results.

with DDPM sampling process, *i.e.*, for $t = T \rightarrow 1$:

$$\mathbf{x}_{t-1} = \frac{1}{\sqrt{\alpha_t}} \left( \mathbf{x}_t - \frac{1 - \alpha_t}{\sqrt{1 - \bar{\alpha}_t}} \epsilon_\phi(\mathbf{x}_t; y, t) \right) + \sigma_t \boldsymbol{\epsilon}, \tag{4}$$

where $\boldsymbol{\epsilon} \sim \mathcal{N}(\mathbf{0}, \mathbf{I})$, $\alpha_t$ is training noise schedule and $\sigma_t$ is noise variance, *e.g.*, $\sqrt{1 - \alpha_t}$.

Note that for SDS training, $t$ is randomly sampled as shown in Eqn. 3 with *red* color, but for DDPM sampling, $t$ is strictly ordered for Eqn. 4 as highlighted in *blue*. Since diffusion model is a general denoiser best utilized by iteratively transforming noisy content to less noisy ones, we argue that random timestep sampling in the optimization process of 3D model is ***unaligned*** with the sampling process in DDPM.

**Supervision Misalignment.** For diffusion models, the denoising prediction provides different granularity of information at different timestep $t$: from coarse structure to fine details as $t$ decreases. To demonstrate this, we visualize the update gradient $\|\boldsymbol{\epsilon}_\phi(\mathbf{x}_t; y, t) - \boldsymbol{\epsilon}\|$ for 3D NeRF model renderings in Figure 3. From the left visualization, it is evident that as NeRF optimization progresses, the diffusion timestep that is most informative to NeRF update changes (as highlighted by curved arrow in Figure 3). We provide more examples to reveal the same pattern on the right. Uniform $t$ sampling disregards the fact that different NeRF training stages require different granularity of supervision. Such misalignment leads to ineffective and inaccurate supervision from SDS in the training process, leading to slower convergence and lower-quality generations (lack of fine details), respectively.

**Out-of-distribution (OOD).** For pre-trained $\epsilon_\phi$ to provide a well-informed score estimation, $\mathbf{x}_t$ needs to be close to the training data distribution which are noised natural images. However, at the early training stages, the renderings of NeRF are obviously out-of-distribution (OOD) to pre-trained Stable Diffusion. Evident frequency difference exists between the rendered images and diffusion model's training data, as shown in Figure 4. We further show in Figure 5 with 2D examples that the lack of high-frequency signal at early stage of content creation directly contributes to mode collapse (low-diversity models given the same prompt) as observed in Poole et al. (2022).

### 3.3 TIME PRIORITIZED SCORE DISTILLATION

Drawbacks of uniform $t$-sampling in vanilla SDS motivate us to sample $t$ more effectively. Intuitively, **non-increasing $t$-sampling** (marked by the curved arrow in Figure 3) is more effective to the 3D optimization process, since it provides coarse-to-fine supervision signals which are more informative to the generation process, and initiates with large noise to avoid the OOD problem caused by low-frequency 3D renderings.

Based on this observation, we first try a naive strategy that decreases $t$ linearly with optimization iteration. However, it fails with severe artifacts in the final rendered image, as shown in App. Fig. 13. We observe that decreasing $t$ works well until later optimization stage when small $t$ dominates. We visualize the SDS gradients (lower-right box within each rendered image) and notice that at small $t$, variance of the SDS gradients are extremely high, which makes convergence difficult for 3D-consistent representation. In fact, different denoising $t$ contributes differently (Choi et al., 2022) to content generation, so it is non-optimal to adopt a uniformly decreasing $t$. In result, we propose a weighted non-increasing $t$-sampling strategy for SDS.

**Weighted non-increasing $t$-sampling** aims to modulate the timestep descent process based on a given normalized weight function $W(t)$. Specifically, $W(t)$ represents the importance of a diffusion timestep $t$ and controls its decreasing speed. For example, a large value of $W(t)$ corresponds to a flat decrease, while a small one corresponds to a steep decline.

To sample the timestep $t(i)$ of the current iteration step $i$, weighted non-increasing $t$-sampling can be easily implemented by:

$$t(i) = \arg \min_{t'} \left| \sum_{t=t'}^{T} W(t) - i/N \right|, \tag{5}$$

where $N$ represents the maximum number of iteration steps, and $T$ denotes the maximum training timestep of the utilized diffusion model.

**Prior weight function** $W(t)$ cannot be trivially derived. As shown in App. Fig. 13, a naively designed constant $W(t)$ (*i.e.*, $t$ decreases linearly) causes training to diverge. To this end, we propose a carefully designed weight function of the form $W(t) = \frac{1}{Z} W_d(t) \cdot W_p(t)$, where $W_d(t)$ and $W_p(t)$ respectively takes into account the characteristics of the diffusion training and the 3D generation process, and $Z = \sum_{t=1}^{T} W_d(t) \cdot W_p(t)$ is the normalizing constant.

- For $W_d(t)$, we derive it from an explicit form of the SDS loss:

$$\mathcal{L}_{\text{SDS}}(\phi, \mathbf{x}) = \mathbb{E}_{t, \epsilon} \left[ \frac{1}{2} w'(t) \left\| \boldsymbol{x} - \text{stop\_grad}(\hat{\boldsymbol{x}}_0) \right\|_2^2 \right], \tag{6}$$

  where

$$\hat{\boldsymbol{x}}_0 = \left( \boldsymbol{x}_t - \sqrt{1 - \bar{\alpha}_t} \boldsymbol{\epsilon}_\phi(\mathbf{x}_t; y, t) \right) / \sqrt{\bar{\alpha}_t} \tag{7}$$

  is the estimated original image, and the SDS loss can be seen as a weighted image regression loss. For simplicity, we start our derivation for $W_d(t)$ by setting $w'(t) = 1$ in Eqn. 6. Then, by substituting Eqn. 1 and Eqn. 7 to Eqn. 6, we can reduce Eqn. 6 to Eqn. 3, with $w(t) = \sqrt{\frac{1 - \bar{\alpha}_t}{\bar{\alpha}_t}}$, which we naturally set as $W_d(t)$. Importantly, the term $\sqrt{\frac{1 - \bar{\alpha}_t}{\bar{\alpha}_t}}$ is also the square root for reciprocal of a diffusion model's signal-to-noise ratio (SNR) (Lin et al., 2023), which characterizes the training process of a diffusion model.

- For $W_p(t)$, we notice that denoising with different timesteps (*w.r.t.* noise levels) focuses on the restoration of different visual concepts (Choi et al., 2022). When $t$ is large, the gradients provided by SDS concentrate on coarse geometry, while $t$ is small, the supervision is on fine details which tends to induce high gradients variance. Thereby we intentionally assign less weights to such stages, to focus our model on the informative content generation stage when $t$ is not in the extreme. For simplicity, we formulate $W_p(t)$ to be a simple Gaussian probability density function, namely, $W_p(t) = e^{-\frac{(t-m)^2}{2s^2}}$, where $m$ and $s$ are hyper-parameters controlling the relative importance of the three stages illustrated in Fig. 6 (b). Analysis on the hyper-parameters $m$ and $s$ is presented in App. C. The intuition behind our $W_p$ formulation is that a large diffusion timestep $t$ induces gradients that are of low-variance but rather coarse, lacking necessary information on details,

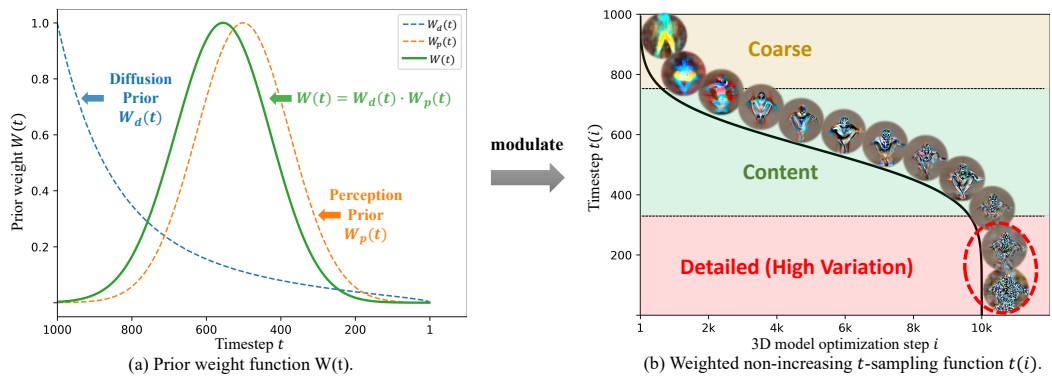

Figure 6: The proposed prior weight function $W(t)$ to modulate the non-increasing $t$-sampling process $t(i)$ for score distillation, as described in Eqn. 5. Weight functions $W(t)$, $W_d(t)$, and $W_p(t)$ are normalized to 0-1 for best visualization. Notice that $W_d$ skews $W$ to avoid small diffusion timesteps that induce high-variance gradients. Such steps and their induced gradient variance are illustrated in (b) as the *Detailed* stage.

while a small $t$ highly raises gradient variance that can be detrimental to model training. Thus we employ a bell-shaped $W_p$ so as to suppress model training on extreme diffusion timesteps.

- To sum up, our normalized weight function $W(t)$ is:

$$W(t) = \frac{1}{Z} \cdot \sqrt{\frac{1 - \bar{\alpha}_t}{\bar{\alpha}_t}} e^{-\frac{(t-m)^2}{2s^2}}. \tag{8}$$

**Time Prioritized SDS.** To summarize, we present the algorithm for our Time Prioritized SDS (TP-SDS) in Alg. 1. We also illustrate the functions $W_d$, $W_p$ and $W$ in Figure 6 (a), where one should notice that $W_d$ skews $W$ to put less weight on the high-variance stage, where $t$ is small. The modulated monotonically non-increasing timestep sampling function is illustrated in Fig. 6 (b).

**Discussion.** The merits of our method include:

- The prior weight function $W(t)$ assigns less weight to those iteration steps with small $t$, thereby avoiding the training crash caused by high variance in these steps.

- Instead of directly adjusting $w(t)$ from Eqn. 3 to assign various weights to different iteration steps, which can hardly affect the 3D generation process (Poole et al., 2022), TP-SDS modifies the decreasing speed of $t$ in accordance with the sampling process of diffusion models. This way, the updates from those informative and low-variance timesteps dominate the overall updates of TP-SDS, making our method more effective in shaping the 3D generation process.

## 4 EXPERIMENT

We conduct experiments on the generation of 2D images and 3D assets for a comprehensive evaluation of the proposed time prioritized score distillation sampling (TP-SDS). For 2D experiments, the generator $g$ (recall definitions in Sec. 3.1) is an identity mapping while $\theta$ is an image representation. For 3D experiments, $g$ is a differentiable volume renderer that transforms 3D model parameters $\theta$ into images.

### 4.1 FASTER CONVERGENCE

We empirically find that the proposed non-increasing $t$-sampling strategy leads to a faster convergence, requiring $\sim 75\%$ fewer optimization steps than the uniform $t$-sampling. This is likely due to more efficient utilization of information, *e.g.*, it is wasteful to seek structure information at later stage of optimization when the 3D model is already in good shape. We conduct both qualitative and quantitative evaluation to demonstrate the fast convergence of our TP-SDS.

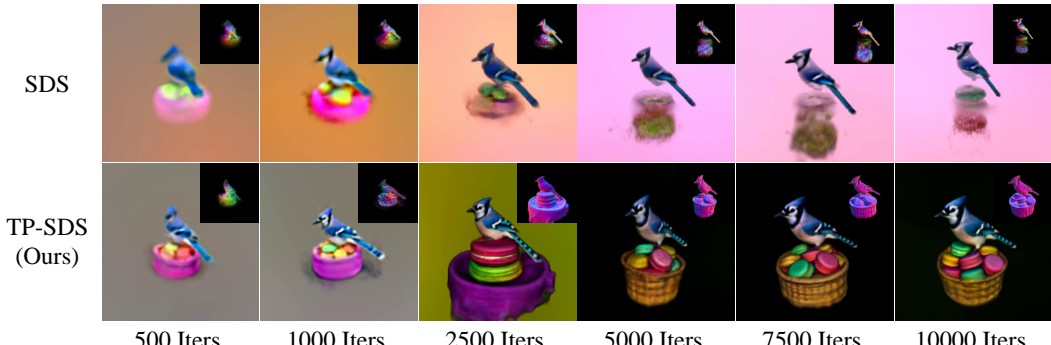

|  | 500 Iters. | 1000 Iters. | 2500 Iters. | 5000 Iters. | 7500 Iters. | 10000 Iters. |

Figure 7: Faster Convergence. Qualitative comparisons of the SDS baseline (first row in each example) and the proposed TP-SDS (second row in each example) under different iteration steps (from 500 to 10000). The proposed TP-SDS leads to faster content generation than the SDS baseline.

**Qualitative comparison.** Figure 7 shows the 3D generation results with different max iteration steps, using the vanilla SDS and our TP-SDS. It is clear that with TP-SDS, the emergence of content (*e.g.*, object structures) is faster with better appearance and details.

**Quantitative evaluation.** Given the 153 text prompts from object-centric COCO validation set, we show in Figure 8 the R-Precision scores of 2D generation results at different iteration steps using the vanilla SDS and our TP-SDS. The growth rate of TP-SDS curves is consistently higher across various CLIP models, which implies a faster convergence requiring significantly fewer optimization steps to reach the same R-Precision score. This leads to the production of superior text-aligned generations at a quicker pace with fewer resources.

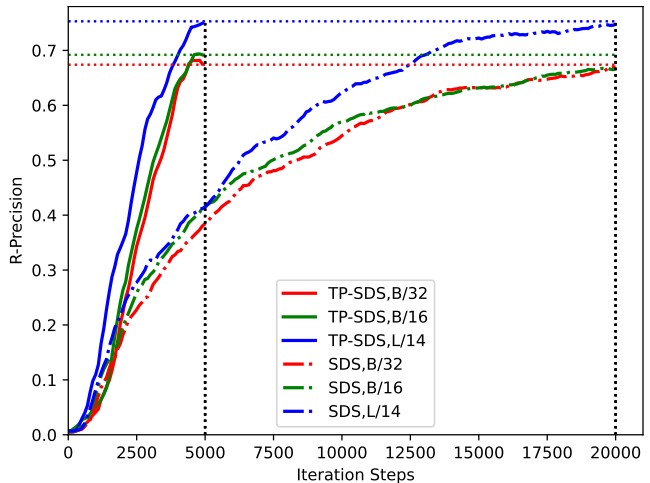

Figure 8: R-Precision curves of 2D generation results produced by SDS and our TP-SDS, given 153 text prompts from object-centric COCO validation set. We use three CLIP models: B/32, B/16, and L/14, for R-Precision evaluation. The R-Precision for TP-SDS have a steeper growth rate compared to those of SDS, signifying faster convergence.

## 4.2 BETTER QUALITY

In this subsection, we demonstrate the effectiveness of TP-SDS in improving visual quality. We argue that some challenging problems in text-to-3D generation, such as unsatisfactory geometry, degenerate texture, and failure to capture text semantics (attribute missing), can be effectively alleviated by simply modifying the sampling strategy of timestep $t$.

**Qualitative comparisons.** We compare our method with the SDS (Poole et al., 2022) baseline, which is a DeepFloyd (Saharia et al., 2022)-based DreamFusion implementation using publicly-accessible threestudio codebase (Guo et al., 2023).

- **More accurate semantics.** The contents highlighted in orange in Figure 9 shows that our generations align better with given texts and are void of the attribute missing problem. For example, SDS fails to generate the mantis' roller skates described by the text prompt as TP-SDS does.
- **Better generation shape and details.** In Figure 9, the contents highlighted in red and green respectively demonstrate that TP-SDS generates 3D assets with better geometry and details. It noticeably alleviates shape distortion and compromised details. For example, in contrast to SDS, TP-SDS successfully generates a robot dinosaur with the desired geometry and textures.
- **Widely applicable.** As shown in Fig. 1, the proposed TP-SDS is highly general and can be readily applicable to various 3D generation tasks, such as text-to-3D scene generation, text-to-3D avatar generation, etc., to further improve the generation quality.

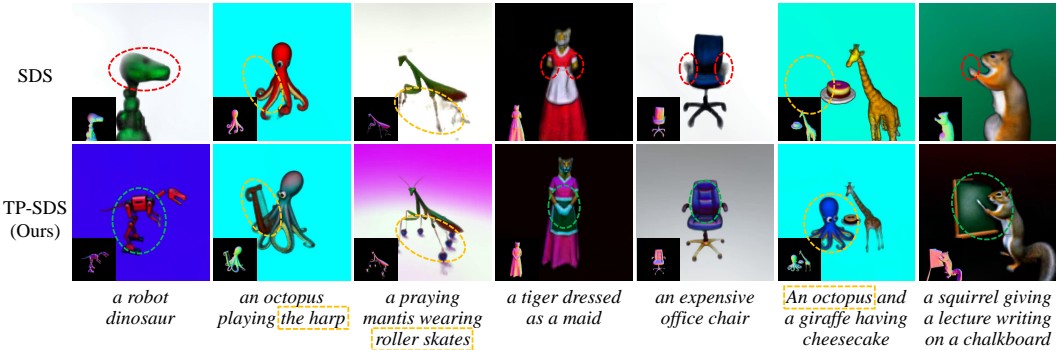

Figure 9: Qualitative comparisons between the original SDS (Poole et al., 2022) (upper row) and TP-SDS (lower row). Our method alleviates the problems of attribute missing, unsatisfactory geometry, and compromised object details at the same time, as highlighted by the colored circles.

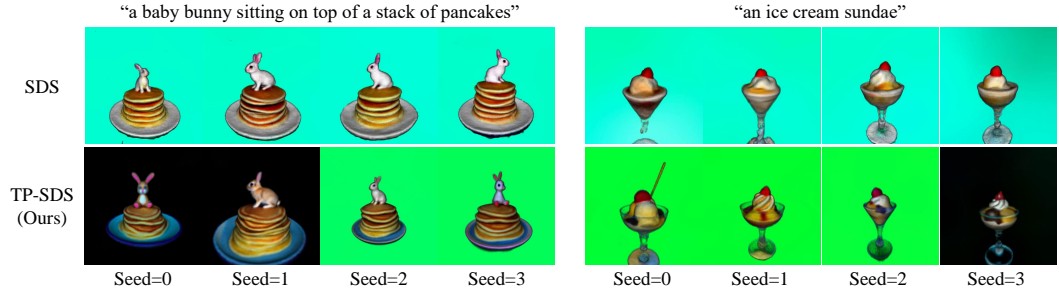

Figure 10: **Higher Diversity.** We compare the diversity of text-to-3D generations between TP-SDS and SDS. The text prompts are provided in the figure. Given different random seeds, TP-SDS is able to generate visually distinct objects, while the results produced by DreamFusion all look alike.

### 4.3 HIGHER DIVERSITY

A key ingredient of AIGC is diversity. In 2D scenarios, given a text prompt, Stable Diffusion is able to generate a countless number of diverse samples while respecting the given prompt. However, the success has not yet been transplanted to its 3D counterpart. Current text-to-3D generation approaches reportedly suffer from the mode collapse problem (Poole et al., 2022), *i.e.*, highly similar 3D assets are always yielded for the same text prompt whatever the random seed is. In Figure 5, we illustrate with 2D generations that the problem of mode collapse is largely caused by the low-frequency nature of initial NeRF renderings, and that the proposed TP-SDS is able to circumvent it by applying large noise (*i.e.*, with a large $t$) during the early training process. Figure 10 further demonstrates that the 3D generations produced by TP-SDS are much more diverse visually than those from DreamFusion (Poole et al., 2022).

## 5 CONCLUSION

**Conclusion.** We propose DreamTime, an improved optimization strategy for diffusion-guided content generation. We thoroughly investigate how the 3D formation process utilizes supervision from pre-trained diffusion models at different noise levels and analyze the drawbacks of commonly used score distillation sampling (SDS). We then propose a non-increasing time sampling strategy (TP-SDS) which effectively aligns the training process of a 3D model parameterized by differentiable 3D representations, and sampling process of DDPM. With extensive qualitative comparisons and quantitative evaluations we show that TP-SDS significantly improves the convergence speed, quality and diversity of diffusion-guided 3D generation, and considerably more preferable compared to accessible 3D generators across different design choices such as foundation generative models and 3D representations. We hope that with DreamTime, 3D content creation can be more accessible for creativity and aesthetics expression.

**Social Impact.** Social impact follows prior 3D generative works such as DreamFusion. Due to our utilization of Stable Diffusion (SD) as the 2D generative prior, TP-SDS could potentially inherit the social biases inherent in the training data of SD. However, our model can also advance 3D-related industries such as 3D games and virtual reality.

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

# A  IMPLEMENTATION DETAILS

Our implementation are based on the publicly-accessible threestudio codebase (Guo et al., 2023), and only the timestep sampling strategy is modified. We provide the algorithm flow of TP-SDS as shown in Alg. 1.

---

**Algorithm 1:** Time Prioritized SDS (TP-SDS).

---

**Input:** A differentiable generator $g$ with initial parameters $\boldsymbol{\theta}_0$ and number of iteration steps $N$,
pre-trained diffusion model $\phi$, prior weight function $W(t)$, learning rate lr, and text prompt $y$.

1 **for** $i = 1, ..., N$ **do**
2     $t(i) = \arg\min_{t'} \left| \sum_{t=t'}^{T} W(t) - i/N \right|$;
3     $\boldsymbol{\theta}_i = \boldsymbol{\theta}_{i-1} - \text{lr} \cdot \nabla_{\boldsymbol{\theta}} \mathcal{L}_{\text{SDS}}(\phi, g(\boldsymbol{\theta}_{i-1}); y, t(i))$;
4 **end**
**Output:** $\theta_N$.

---

# B  ABLATION STUDY

In this section we evaluate the 3D generation ability of TP-SDS by formulating $W(t)$ as various functions, to investigate the effects of the prior weight function (recall that $T$ denotes the maximum training timestep of the utilized diffusion model):

- **Baseline**: We provide the results produced by the random timestep schedule for reference.

- **Linear**: $W(t) = \frac{1}{T}$, which induces a linearly decaying time schedule.

- **Truncated linear**: $W(t) = \frac{1}{T}$, but we constrain $t(i)$ to be at least 200, instead of 1. This is for assessing the influence of gradient variance induced by small $t$.

- $W_p$ **only**: $W(t) = W_p(t)$.

- $W_t$ **only**: $W(t) = W_d(t)$.

- **TP-SDS**: $W(t) = W_d(t) \cdot W_p(t)$.

The results are shown in Fig. 11. Evidently, naive designs such as *baseline*, *linear* and *linear (truncated)* fail to generate decent details, where a valuable observation is that *linear (truncated)* generates finer details than that of *linear*, validating our claim that small $t$ hinges 3D generation with high gradient variance. Moreover, applying $W_p$ or $W_d$ alone cannot produce satisfying 3D results, suffering either over-saturation or loss of geometrical details. In comparison, our proposed prior weight function circumvents all the deficiencies and generates delicate 3D.

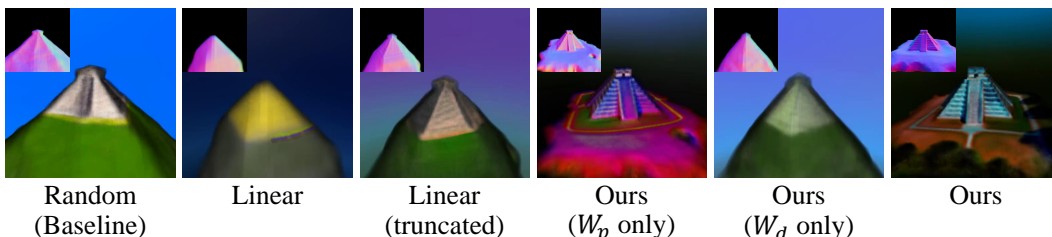

| Random
(Baseline) | Linear | Linear
(truncated) | Ours
($W_p$ only) | Ours
($W_d$ only) | Ours |

Figure 11: Ablation study. We investigate the effects of the prior weight function by modifying its formulation and comparing the text-to-3D generation results. The text is "Chichen Itza, aerial view".

## C HYPER-PARAMETER ANALYSIS

TP-SDS improves generation efficiency, quality, and diversity compared to the SDS baseline. However, the proposed prior weight function $W(t)$ parameterized by $\{m, s\}$ introduces extra hyper-parameters. Thus we explore the influence of these hyper-parameters on text-to-3D generation, which can serve as a guide for tuning in practice. In Figure 12, we illustrate the impact of hyper-parameters on the generated results in a large search space. Specifically, a large $s$ significantly increases the number of optimization steps spent on both the *coarse* and the high-variance *detailed* stages, degrading the generation quality. $m$ controls the model to concentrate on different diffusion timesteps, and a rule of thumb is to make $m$ close to $\frac{T}{2}$.

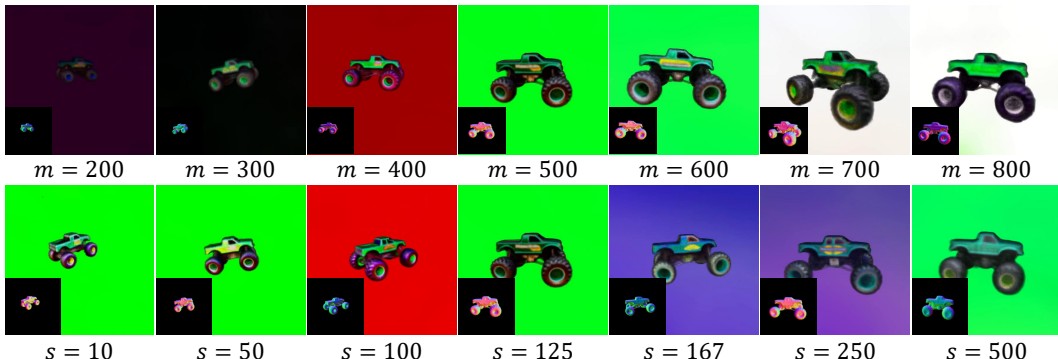

Figure 12: Influence of the time prior configuration $\{m, s\}$ on 3D generations. The text prompt is "a DSLR photo of a green monster truck". $s$ is set to 125 for the experiments on the first row while $m = 500$ for the rest.

## D TIMESTEP ANALYSIS

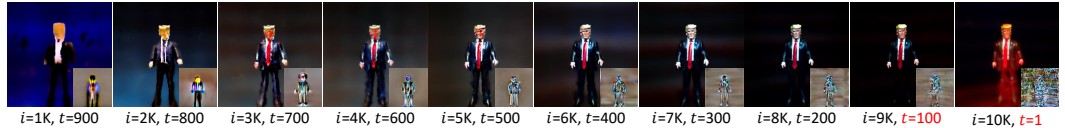

Figure 13: Visualization of NeRF optimization with SDS using a naive $t$-sampling strategy, where $t$ decreases linearly with the iteration step $i$. Severe artifacts appear in the final rendered image due to large gradients variance with small $t$.

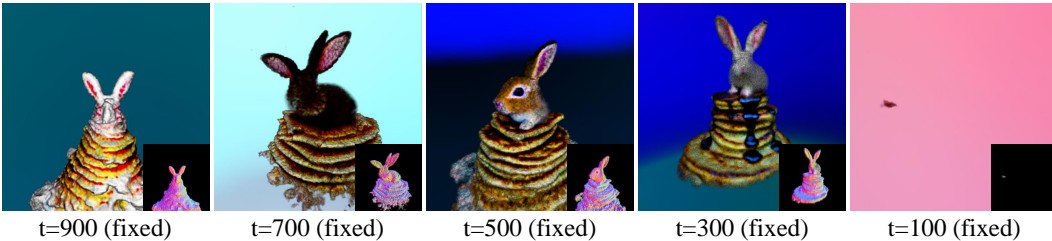

Figure 14: Illustration of information capacity for different diffusion timesteps. Here we train 3D assets supervised by SDS with the diffusion timestep $t$ fixed throughout. Notably, for small $t$, *e.g.*, 100, the gradient variance becomes too high for the model to generate a 3D successfully, while a large $t$ like 900 makes the generation lack local details. Only those $t$ not in the extreme can prompt SDS to produce decent 3D generations, for which we believe that such timesteps are most informative and consequently make our model training concentrate on the "content" stage consisting of such timesteps. The text prompt is "a DSLR photo of a baby bunny sitting on a pile of pancakes".

# E  QUANTITATIVE EVALUATION ON DIFFERENT METHODS

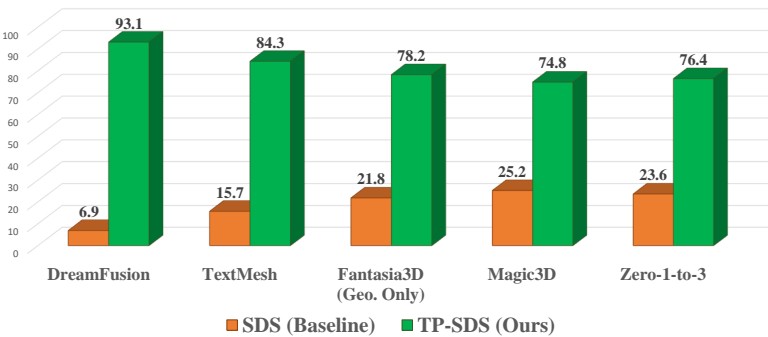

Figure 15: User preference studies of SDS and our TP-SDS on five popular 3D generation methods: DreamFusion (Poole et al., 2022), TextMesh (Tsalicoglou et al., 2023), Fantasia3D (Chen et al., 2023), Magic3D (Lin et al., 2022), and Zero-1-to-3 (Liu et al., 2023a). The implementation of these methods follows threestudio (Guo et al., 2023). We use 415, 100, 100, 100, and 50 prompts as evaluation sets respectively for the above five methods, each evaluated by 10 participants. Our proposed TP-SDS consistently achieves higher preference scores (%).

# F  COMPARISONS WITH TIMESTEP SCHEDULES PROPOSED IN HIFA AND PROLIFICDREAMER

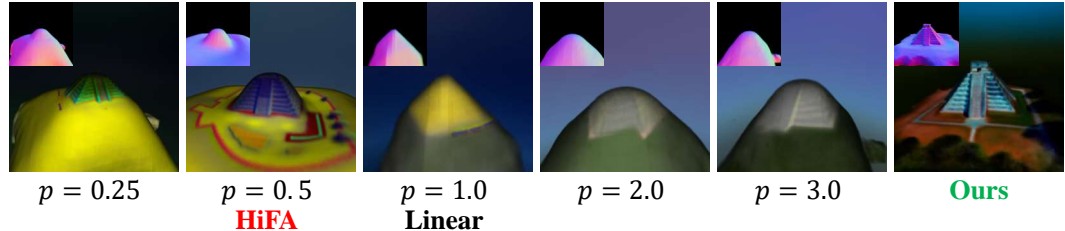

$p = 0.25$    $p = 0.5$    $p = 1.0$    $p = 2.0$    $p = 3.0$    **Ours**
**HiFA**      **Linear**

(a) Comparison of our method with the timestep schedule proposed in HiFA (Zhu & Zhuang, 2023), which adopts an exponential $t$-annealing strategy with a power of 0.5. The results of HiFA variants with different powers (denoted in $p$) are also provided for comprehensive comparison.

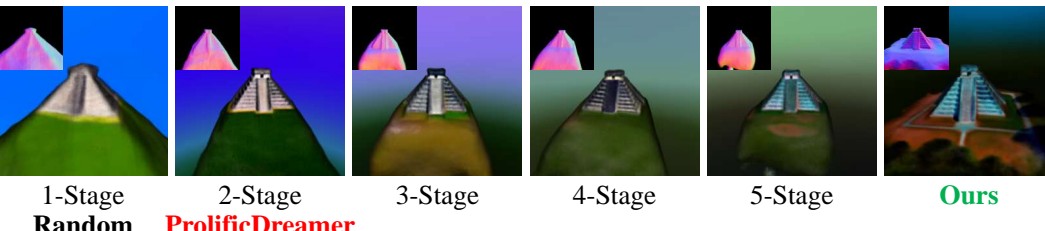

1-Stage    2-Stage    3-Stage    4-Stage    5-Stage    **Ours**
**Random**  **ProlificDreamer**

(b) Comparison of our method with the timestep schedule proposed in ProlificDreamer (Wang et al., 2023), which adopts a two-stage random $t$-sampling strategy. The results of ProlificDreamer variants with different number of stages are also provided for comprehensive comparison.

Figure 16: Comparisons of our method with the timestep schedules proposed in HiFA (Zhu & Zhuang, 2023) and ProlificDreamer (Wang et al., 2023).

# G  IMPLEMENTATION DETAILS AND MORE RESULTS OF TP-VSD

To produce the VSD-based results presented in Figure 1 and Figure 17, we follow the Prolific-Dreamer (Wang et al., 2023) implementation by threestudio (Guo et al., 2023) and modify its timestep schedule. Specifically, we adopt the random uniform $t$-sampling for VSD, and use our proposed timestep schedule for TP-VSD. We emphasize that we adopt the same hyper-parameter configuration $\{m = 500, s = 125\}$ for both TP-SDS and TP-VSD throughout this paper. Further tuning of hyper-parameters might yield better results.

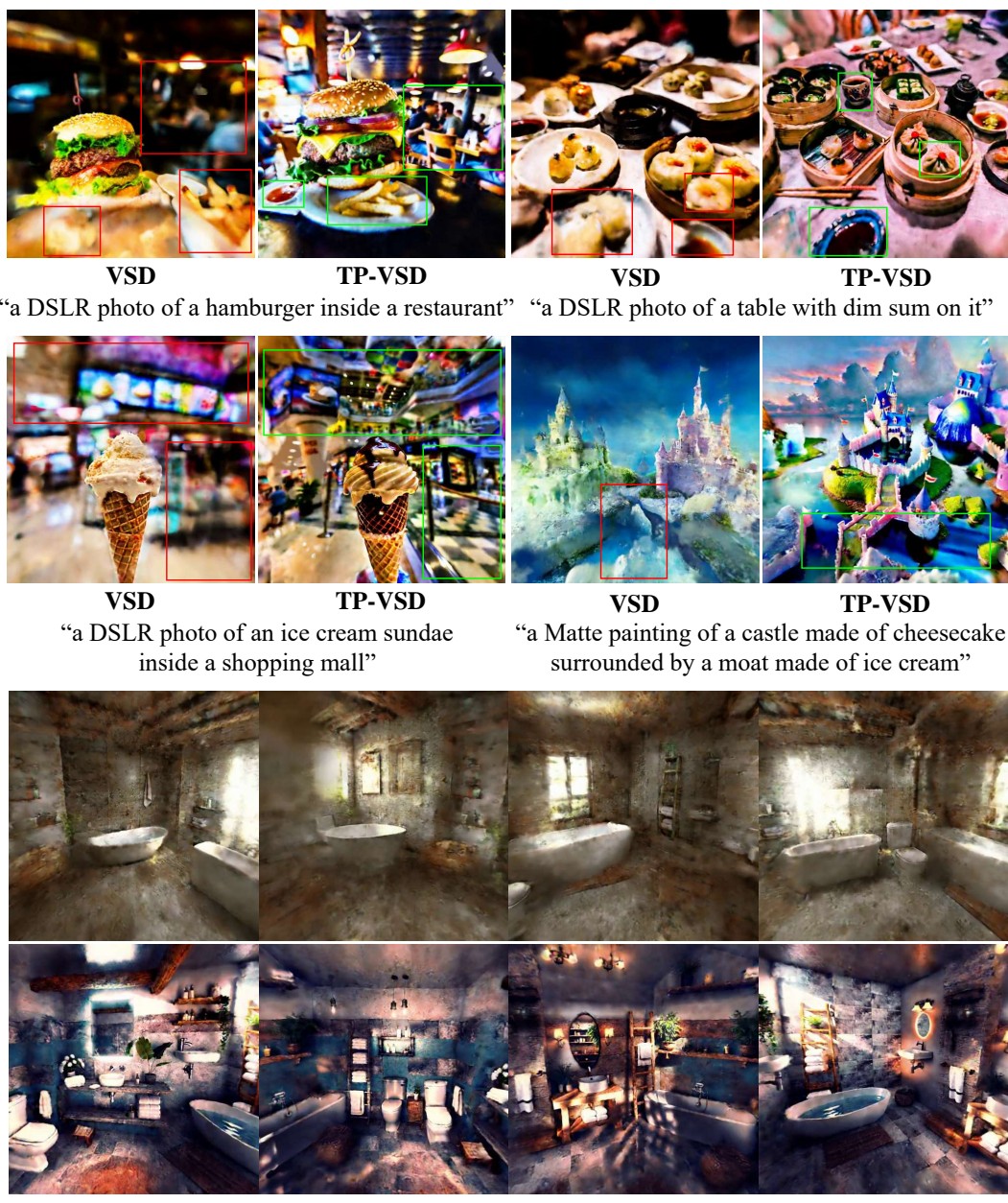

**VSD**     **TP-VSD**     **VSD**     **TP-VSD**

"a DSLR photo of a hamburger inside a restaurant"   "a DSLR photo of a table with dim sum on it"

**VSD**     **TP-VSD**     **VSD**     **TP-VSD**

"a DSLR photo of an ice cream sundae
inside a shopping mall"   "a Matte painting of a castle made of cheesecake
surrounded by a moat made of ice cream"

"bathroom, rustic style, realistic, 4k" (Top: **VSD**, Bottom: **TP-VSD**)

Figure 17: More examples of VSD (Wang et al., 2023) and our proposed TP-VSD. Our method is better at generating details and improving clarity, as highlighted with red and green bounding boxes.

# H EFFECTIVENESS ON DIFFERENT 3D REPRESENTATIONS

Our proposed $t$-annealing schedule is an improvement on the diffusion-guided optimization process and therefore is universally applicable to various 3D representations, such as NeRF (Mildenhall et al., 2021), NeuS (Wang et al., 2021), and DMTet (Shen et al., 2021).

Both quantitative and qualitative evaluations demonstrate the effectiveness of our method on different 3D representations. For quantitative evaluations in Figure 15, DreamFusion and Zero-1-to-3 are NeRF-based methods, TextMesh is a NeuS-based method, Fantasia3D is a DMTet-based method, Magic3D is a hybrid (NeRF + DMTet) method. For qualitative evaluations, we provide NeRF-based results in Figure 7, Figure 9, Figure 10, Figure 17, and Figure 21 (the first to third rows), NeuS-based results in Figure 19 and Figure 22, DMTet-based results in Figure 18 and Figure 21 (the fourth row).

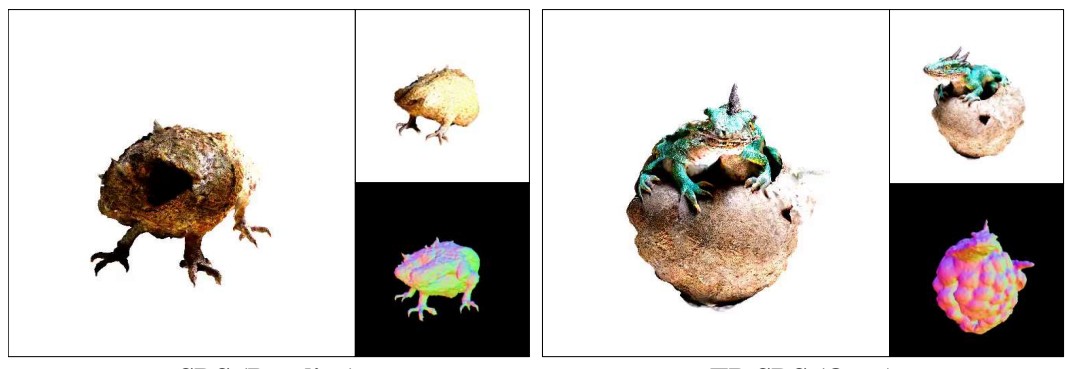

**SDS (Baseline)**          **TP-SDS (Ours)**

"a DSLR photo of a baby dragon hatching out of a stone egg"

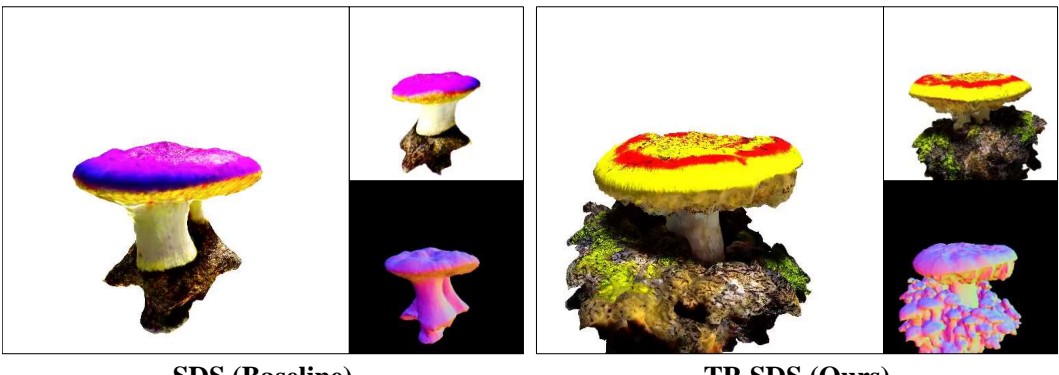

**SDS (Baseline)**          **TP-SDS (Ours)**

"a brightly colored mushroom growing on a log"

Figure 18: Qualitative comparison of SDS and TP-SDS based on the threestudio (Guo et al., 2023) implementation of Fantasia3D (Chen et al., 2023).

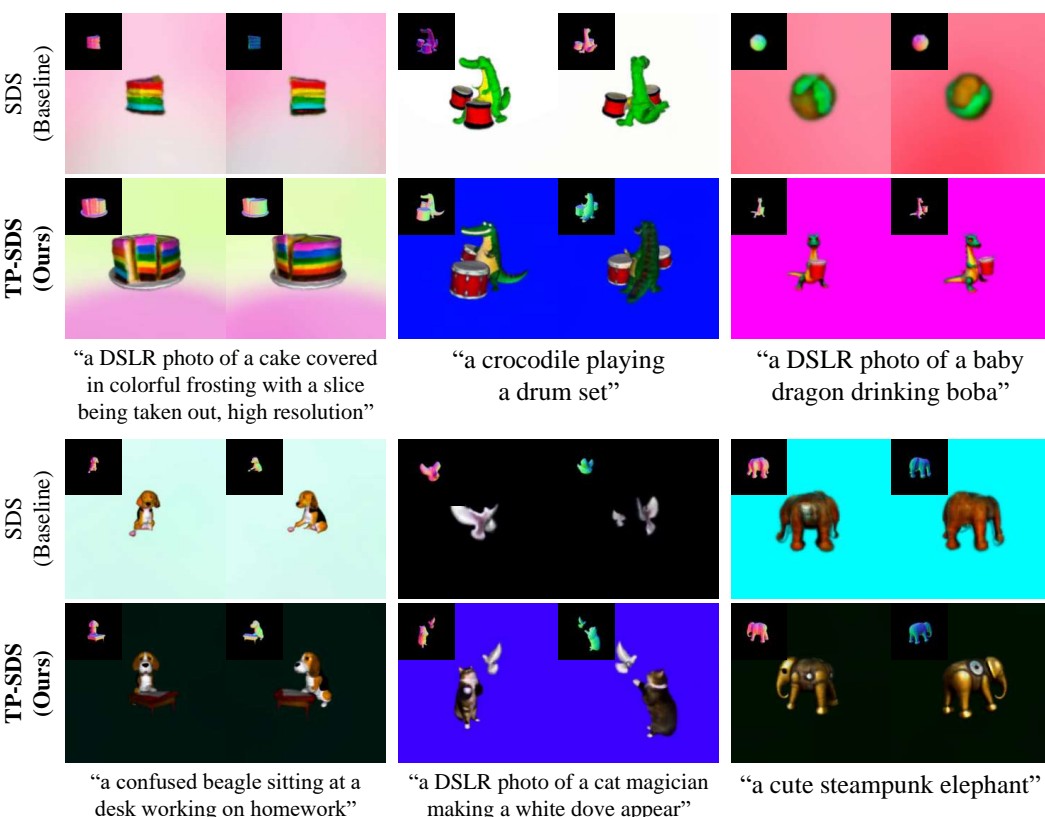

Figure 19: Qualitative comparison of SDS and TP-SDS based on the threestudio (Guo et al., 2023) implementation of TextMesh (Tsalicoglou et al., 2023).

# I VISUALIZATIONS OF SDS-BASED OPTIMIZATION ON NEUS AND WITH INITIALIZED NERF

As a complement to Figure 3, we further provide two cases: (a) NeuS Wang et al. (2021) optimized by DeepFloyd-IF and (b) SMPL-initialized NeRF Mildenhall et al. (2021) optimized by Stable-Diffusion, and visualized the SDS gradients under different timesteps, as shown in Figure 20.

The supervision misalignment issue described in Section 3.2 can also be observed in Figure 20, indicating that using a decreasing timestep schedule remains a good principle. Additionally, for well-initialized NeRF, optimization with large timesteps should be avoided.

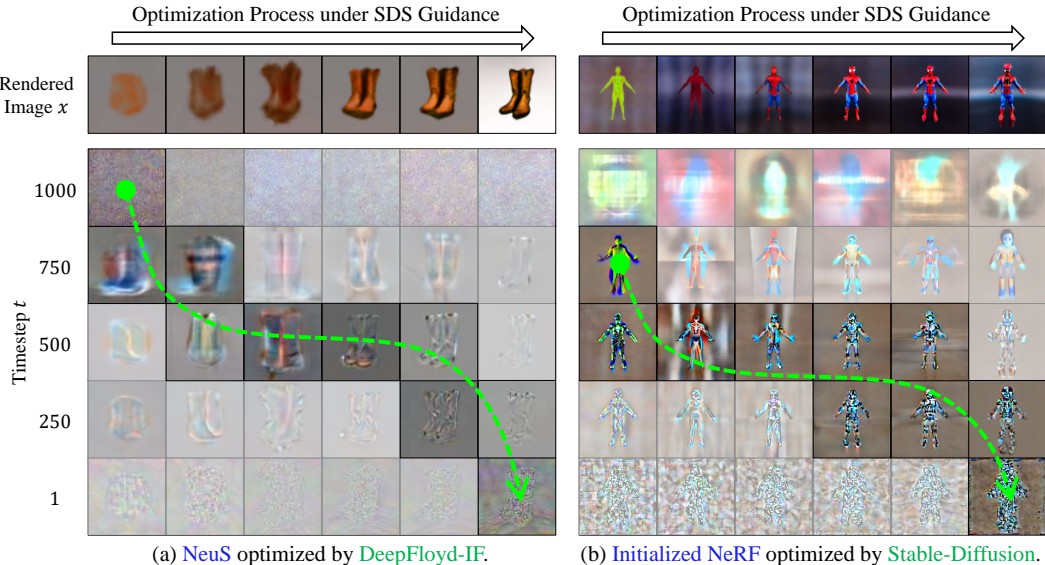

(a) NeuS optimized by DeepFloyd-IF.  (b) Initialized NeRF optimized by Stable-Diffusion.

Figure 20: Visualization of SDS gradients under different timesteps $t$. Two cases: (a) NeuS optimized by DeepFloyd-IF and (b) initialized NeRF optimized by Stable-Diffusion, are provided. In particular, initialized NeRF is pre-trained on the images rendered from SMPL Bogo et al. (2016) mesh template. The supervision misalignment issue described in Section 3.2 can also be observed. We use an imaginary green curved arrow to denote the path for more informative gradient directions as optimization progresses under SDS guidance.

## J  EXAMPLES OF ALLEVIATING BLURRINESS AND COLOR DISTORTION

We provide more examples to show that the proposed timestep schedule can alleviate the issues of blurriness and color distortion, as shown in Figure 21.

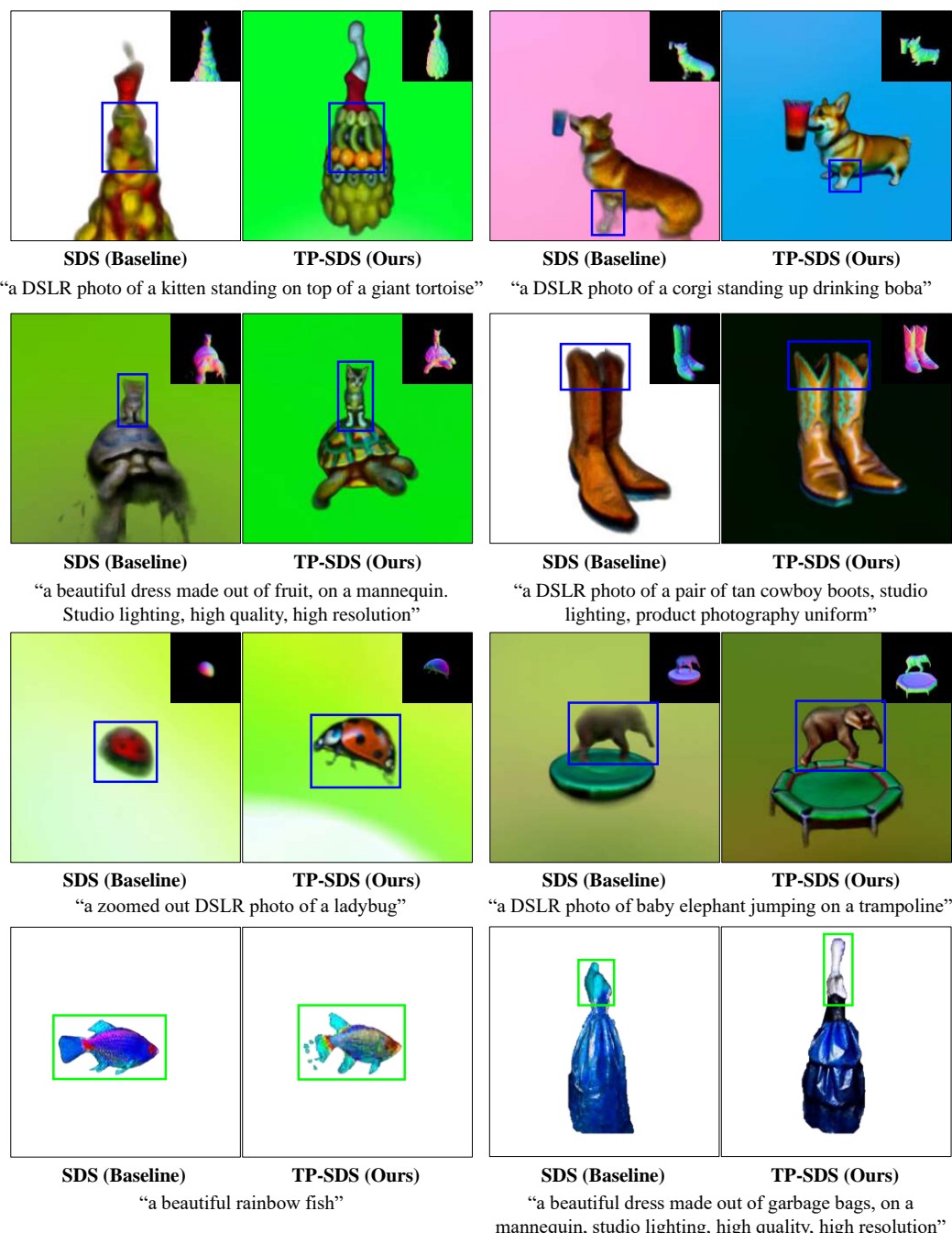

Figure 21: Examples to demonstrate that our method can alleviate blurness and color distortion issues, as highlighted with blue and green bounding boxes, respectively. The first three rows are the results of DreamFusion (Poole et al., 2022), and the fourth row is the results of Magic3D (Lin et al., 2022), implemented by threestudio (Guo et al., 2023).

# K    EFFECTIVENESS ON TEXT-TO-AVATAR GENERATION

In figure 1, we show that our proposed TP-SDS facilitates ControlNet-based text-to-avatar generation work DreamWaltz (Huang et al., 2023), avoiding texture and geometry loss. Here we further demonstrate that another popular text-to-avatar work, AvatarCraft (Jiang et al., 2023), can also benefit from our method, achieving more detailed textures and avoiding color over-saturation, as shown in Figure 22.

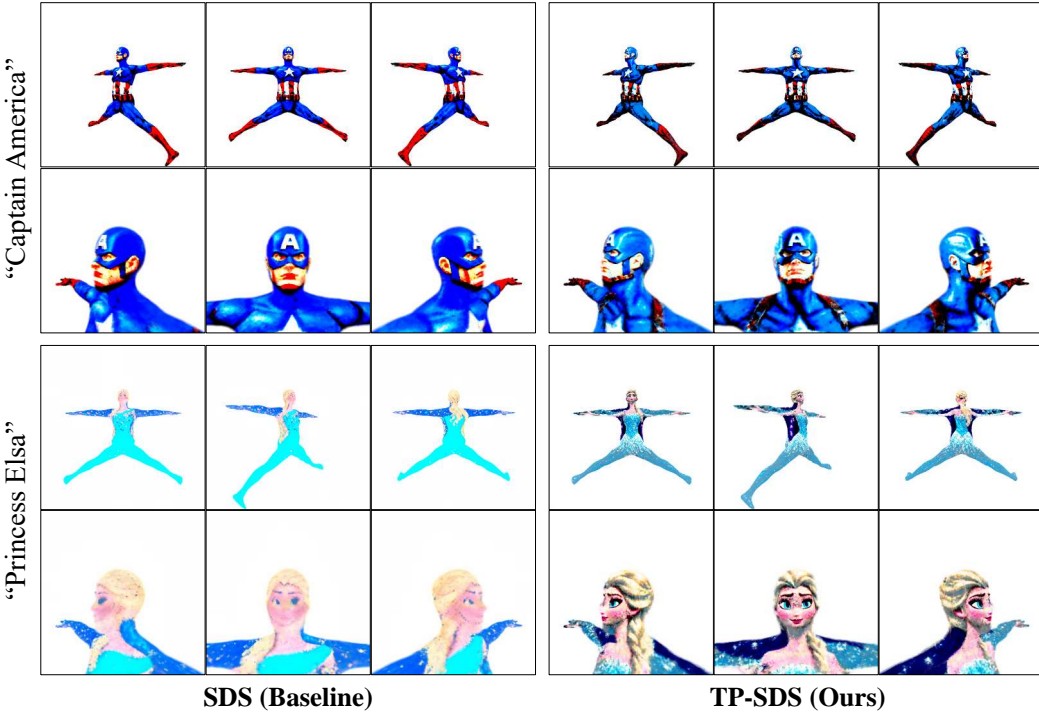

Figure 22: Qualitative comparison of SDS and our TP-SDS based on the popular text-to-3D-avatar generation work, AvatarCraft (Jiang et al., 2023). Compared to the SDS baseline, our method can lead to more detailed faces and avoid color over-saturation.

