# OpenReview forum: "DreamTime: An Improved Optimization Strategy for Diffusion-Guided 3D Generation"
_ICLR.cc/2024/Conference — ICLR 2024 poster_

### Official Review · Reviewer_iLxc · 2023-10-29

**Soundness:** 3 good
**Presentation:** 4 excellent
**Contribution:** 3 good
**Rating:** 8
**Confidence:** 4

**Summary:**

This paper proposes to use a monotonically non-increasing time schedule in the score distillation training for 3D generation. The paper identifies the different denoising effects of different time steps, e.g. larger t helps generate coarse geometry and smaller t helps generate fine texture details. The proposed method shows much faster convergence speed than its random time schedule counterpart, and generates results with more geometric and textural details.

**Strengths:**

1. The method is well motivated. The original SDS loss is derived from the training loss of diffusion models and thus adopts a random time schedule. But during score distillation for 3D generation, the process is more like the inference procedure of diffusion models. The optimized 3D representation goes from coarse to fine, so the time schedule should also be consistent with this.

2. The presentation is great. The paper motivates the method with an easy-to-understand analysis (Section 3.2) and good illustrations (Figure 3, Figure 6).

3. The proposed method is very easy to implement and yields no additional cost, but still achieves impressive results (Figure 8 and Figure 9).

**Weaknesses:**

1. The proposed time schedule seems to be highly heuristic. I do not think the optimal schedule would be the same for all different prompts. If the schedule is designed to be adaptive to the renderings of the current iteration it would be more robust.

2. In Figure 1 the authors also show the results for "TP-VSD", but I cannot find any more explanation for it. Maybe it is obvious to apply the method on VSD, but I still think a small dedicated piece of text explaining it would be good. For example, do you use the same schedule for both SDS and VSD?

**Questions:**

1. Do you have any idea on how to design an adaptive schedule based on the information of the current instance instead of a fixed schedule?

---

> ### Author Response · Authors · 2023-11-22
>
> Thank you very much for the insightful comments and questions.
>
> **W1. Timestep schedule adaptive to the current renderings.**
>
> We totally agree with the reviewer’s comment that the optimal schedule might be different for different prompts. In this work, we aim to provide a general schedule that works reasonably well across most prompts. Designing a schedule to be adaptive to the renderings of the current optimization state is very exciting future work to be explored.
>
> **W2. More explanation of the VSD and TP-VSD results.**
>
> Thank you very much for pointing this out. We now add a dedicated section G in the appendix, explaining the details of TP-VSD with more quantitative examples demonstrating finer details and clarity after applying our proposed time schedule.
>
> **Q1. Any idea on how to design an adaptive schedule.**
>
> Designing an adaptive schedule instead of a fixed one is challenging (in terms of training time efficiency) but a very exciting direction, we can start with a simple adaptive strategy and see how to make it better. A solution inspired by the greedy algorithm is that, based on the rendered image at each iteration, we exhaustively enumerate all timesteps to determine which $t$ can bring the greatest quality or information gain (e.g. measured by CLIP score), and select the optimal one for the current iteration. More delicate design remains to be explored for future work.

---

> > ### Comment · Reviewer_iLxc · 2023-11-23
> >
> > Thanks for the rebuttal. I think this paper is worth accepting and keep my original rating.

---

### Official Review · Reviewer_5Rpq · 2023-10-30

**Soundness:** 3 good
**Presentation:** 3 good
**Contribution:** 3 good
**Rating:** 6
**Confidence:** 5

**Summary:**

Recent advancements in text-to-image diffusion models, trained on vast amounts of image-text pairs, have paved the way for 3D content creation by optimizing an initially random differentiable 3D representation using score distillation. Despite its promise, the method has substantial drawbacks: a prolonged optimization convergence and subpar 3D model outcomes in terms of both quality (e.g., distorted features and textures) and diversity, especially when juxtaposed with text-guided image synthesis. This paper pinpoints the crux of the problem in the misalignment between the 3D optimization and the uniform timestep sampling in score distillation. A novel approach is introduced that reprioritizes timestep sampling using monotonically non-increasing functions, seamlessly integrating the 3D optimization and diffusion model sampling. Comprehensive experiments validate the efficacy of this redesign, showcasing marked improvements in convergence speed, model quality, and diversity.

**Strengths:**

While the paper's methodology is straightforward, its experiments are illuminating. It provides a visualization of NeRF's training steps guided by the SDS loss, uncovering a refined pathway to mitigate the training diversity and challenges associated with SDS loss. Furthermore, the study introduces a weighted function to fine-tune the significance of the t-sampling process for SDS. This not only accelerates convergence but also yields results with enhanced diversity. Moreover, this approach promotes improved semantic alignment.

**Weaknesses:**

The quality of the proposed method doesn't appear to significantly surpass that of Magic3D, Dreambooth3d[1], or Fantasia3D[2].

[1] Raj A, Kaza S, Poole B, et al. Dreambooth3d: Subject-driven text-to-3d generation[J]. arXiv preprint arXiv:2303.13508, 2023. [2] Chen R, Chen Y, Jiao N, et al. Fantasia3d: Disentangling geometry and appearance for high-quality text-to-3d content creation[J]. arXiv preprint arXiv:2303.13873, 2023.

The method introduced seems to be a versatile technique applicable to various 3D representations. Comprehensive evaluations on diverse representations such as NeRF, NeuS, and DMTet should be undertaken.

**Questions:**

In Fig.3, might it be beneficial to switch out NeRF for SDS-based implicit fields such as NeuS or VolSDF? Based on my experiments using CLIP guidance, each representation yields distinct visualizations as optimization unfolds. It raises the question: would these representations evolve similarly under SDS guidance? Occasionally, there might be a scenario where we initialize the neural implicit fields with a rudimentary object. How would the training process evolve in such instances?

The paper posits its objective as refining a differentiable 3D representation by leveraging knowledge distilled from either the pre-trained Stable Diffusion or Zero123, depending on whether the prompt is text or image-based. Could you clarify which outcomes are derived from the SDS loss in tandem with the pre-trained Stable Diffusion and which with Zero123? Additionally, I seem unable to locate results corresponding to image inputs.

While Fig.2 highlights issues of blurriness and color distortion associated with the SDS loss in the context of texture generation, it appears that this paper falls short of presenting compelling evidence to demonstrate that the introduced method effectively addresses both challenges.

Missing citations for AvatarCraft: Transforming Text into Neural Human Avatars with Parameterized Shape and Pose Control, which produces human avatars with SDS guidance. I'm intrigued to know if the methodology presented can be seamlessly adapted for avatar creation.

---

> ### Author Response · Authors · 2023-11-22
>
> Thank you very much for the constructive questions. We’ve provided more results according to your comments in the new revision.
>
> **W1. Limited quality compared to Magic3D, DreamBooth3d, or Fantasia3D.**
>
> We acknowledge that the quality of the results presented in the paper is limited because most results are based on a basic DreamFusion implementation (64x64 NeRF), without using more advanced 3D representations or higher resolutions as in the original papers.
>
> As the proposed timestep schedule improves over general score distillation, our method is orthogonal to these works and can further improve their quality. To demonstrate this, we conduct more evaluations based on Magic3D and Fantasia3D, including quantitative and qualitative results, as shown in Figure 15, Figure 18, and Figure 21 (the fourth row) in the appendix. DreamBooth3D is not included because it is not open-sourced yet. Specifically, in Figure 18, our method improves the quality of Fantasia3D by avoiding semantic and geometric defects; in the fourth row of Figure 21, our method improves the quality of Magic3D by correcting color distortion.
>
> **W2. Comprehensive evaluations on diverse 3D representations.**
>
> Thank you for the suggestion. We further conduct comprehensive evaluations of our proposed timestep schedule on different 3D representations, including: NeRF, NeuS, and DMTet.
> - For quantitative evaluations shown in Figure 15, DreamFusion and Zero-1-to-3 are NeRF-based methods, TextMesh is a NeuS-based method, Fantasia3D is a DMTet-based method, Magic3D is a hybrid (NeRF + DMTet) method.
> - For qualitative evaluations, we provide NeRF-based results in Figure 7, Figure 9, Figure 10, Figure 17, and Figure 21 (the first to third rows), NeuS-based results in Figure 19 and Figure 22, DMTet-based results in Figure 18 and Figure 21 (the fourth row).
>
> **Q1. Training processes of different neural implicit fields under SDS guidance.**
>
> Thank you for the question. We follow the reviewer's comment and further unfold the optimization processes of NeuS and initialized NeRF under SDS guidance, and visualize the gradients across different timesteps, as shown in Section I Figure 20.
>
> From the results, we observe that: (a) neural implicit fields follow a coarse-to-fine generation process; (b) gradients at different timesteps provide different granularity of information. Based on the above observations, we can conclude that the supervision misalignment issue described in Section 3.2 also exists in the optimization processes of NeuS and initialized NeRF under SDS guidance. As a result, the proposed $t$-annealing schedule should be applicable to these different neural implicit fields. Specially, for NeRF initialized with a rudimentary object, SDS optimization with large timesteps (w.r.t large noise injection) should be avoided, because it could be detrimental to the initialization prior.
>
>
> **Q2. Clarification of the diffusion models used.**
>
> Our apologies for the unclear presentation, we will revise the manuscript to add information about the diffusion models used. To clarify, only the Zero123 results (in Figure 1 and Figure 15) are derived from the SDS loss in tandem with the image based Zero123 model. The remaining results are based on DreamFusion and TextMesh using the text based DeepFloyd-IF or Stable-Diffusion model.
>
> Most of our experiments are conducted with text prompts instead of image prompts because we believe text-to-3D generation is more fundamental and challenging. Even though our method is applicable to Image-to-3D generation, using pre-trained 3D-aware Zero123 model only requires a few hundred iterations to converge, which makes the analysis and evaluation of timestep schedule less illustrative.
>
>
> **Q3. Evidence to address blurriness and color distortion is lacking.**
>
> Thank you for the the constructive comment. We additionally provide examples to demonstrate the effectiveness of our approach in alleviating blurriness and color distortion issues, as shown in Figure 21.
>
> **Q4. Missing citations for AvatarCraft.**
>
> Thank you for pointing this out. We now add qualitative examples of applying DreamTime to AvatarCraft in Figure 22. We'll add more citations on text-to-avatar generation in later revisions.

---

### Official Review · Reviewer_zkfR · 2023-10-30

**Soundness:** 3 good
**Presentation:** 3 good
**Contribution:** 2 fair
**Rating:** 3
**Confidence:** 4

**Summary:**

The authors propose a method for image-to-3D generation. Specifically, they suggest incorporating depth images during DDIM inversion and sampling to generate view-consistent novel view images. The enhanced novel view images are then used to compute a variant version of SDS loss, i.e., the RGSD loss, to optimize the 3D representation. The results appear to outperform the baselined methods used.

The authors proposed a timestep annealing scheme in SDS to enhance text-to-3D generation. Specifically, they pointed out the suboptimality of the random timestep sampling in Dreamfusion and suggested a non-increasing t-sampling approach using a Gaussian weight function W(t) in SDS. The proposed timestep annealing scheme proves to be effective and outperforms the baselines.

**Strengths:**

The paper is well-written and presents its ideas clearly.

The effectiveness of the proposed timestep annealing scheme in text-to-3D generation using SDS is demonstrated.

**Weaknesses:**

1. There are several existing works that have proposed timestep schedulings, such as two-stage samplings in ProlificDreamer [1] and the non-increasing timestep annealing scheme in HIFA [2]. These methods have been available on arXiv for a while. Given the rapidly evolving nature of this field, even if not officially published, it would be beneficial for the authors to compare the proposed Gaussian PDF with these existing approaches.
2. The enhancement of TP-VSD is intriguing. However, the authors have presented only one example in Fig.1. It is recommended to provide additional visual results, especially for a wider range of objects and scenes as demonstrated in ProlificDreamer.
3. While the proposed improvement is a valuable technique, its contribution as an ICLR paper may be limited.

[1] ****ProlificDreamer: High-Fidelity and Diverse Text-to-3D Generation with Variational Score Distillation, Wang et al., NeurIPS 2023. ****

[2] ****HiFA: High-fidelity Text-to-3D with Advanced Diffusion Guidance, Zhu et al., 2023.****

**Questions:**

I am curious if there exists a concrete theoretical deduction for the timestep annealing scheme that could explain which specific timestep annealing scheme should be employed.

---

> ### Author Response · Authors · 2023-11-22
>
> Thank you very much for the intriguing questions and comments.
>
> **W1. Existing works ProlificDreamer and HiFA have proposed timestep schedules.**
>
> We acknowledge that ProlificDreamer and HiFA are concurrent works to ours, and would like to emphasize that such existing works only treat time schedules as a practical technique, while our work carefully studies the impact of timestep on diffusion-guided 3D generation with detailed analysis.
>
> Nonetheless, we agree with the reviewer's comment and compare our method with the timestep schedules proposed by HiFA and ProlificDreamer, as shown in Section F (Figure 16) in the Appendix. Compared with our results, both competitors fail to generate geometric details. We argue that the reasons for their failures are different:
> - HiFA’s timestep schedule favors training with small t, where high gradient variance leads to color bias and simplified geometry. We also mentioned these effects in Figure 11 and Figure 13. As an exception, the original HiFA paper uses an iterative SDS optimization that may avoid the high variance of small t, which is orthogonal to our work.
> - ProlificDreamer’s timestep schedule could be regarded as a compromise between random t-sampling and linear t-annealing, Consequently, it retains the drawbacks associated with random t-sampling, as discussed in Section 3 of this paper.
>
> **W2. More results of TP-VSD.**
>
> Thank you for the suggestion. We now provide more results of TP-VSD on a wider range of objects and scenes in Section G (Figure 17) in the Appendix.
>
> **W3. Contribution as an ICLR paper may be limited.**
>
> With all due respect, we would like to argue differently. Improving sampling quality and efficiency has been an important subject for diffusion-guided image generation. However, optimization-based 3D generation is known to be time-consuming with sub-optimal generation results, yet understanding and addressing the inefficiencies stemming from the formulation of random t-sampling has been overlooked, which is the focus of our work. We would like to quote reviewer DnQ9 that _"It studies an aspect of using diffusion model for 3D diffusion model -- how to schedule the timestep used to condition the diffusion model, which is not systematically explored in prior literature... The authors also show the generality of this technique by showing results across many state-of-the-art 3D generation methods so it can be easily adopted by the community in any future work along the line."_, reviewer 5Rpq that _"It provides a visualization of NeRF's training steps guided by the SDS loss, uncovering a refined pathway to mitigate the training diversity and challenges associated with SDS loss."_, and reviewer iLxc that _"The method is well motivated."_ We hope this would clarify our contributions more concretely.
>
> **Q1. Concrete theoretical deduction for timestep annealing scheme.**
>
> Thank you for the question. In this paper, we aim to provide a general guideline for timestep sampling, which should a) be monotonically non-increasing for coarse-to-fine generation and to avoid inefficiency b) suppress the sampling of small timesteps to avoid high-variance gradients c) encourage the sampling of medium (content-related) timesteps for most informative guidance.
>
> The proposed timestep annealing scheme is an empirically robust one across prompts, but as pointed out by reviewer iLxc, different prompts may work best under different schedules. While it's challenging to deduct a theoretically optimal schedule to work for all prompts, another possible direction would be to design an adaptive schedule which remains for future work.

---

### Official Review · Reviewer_DnQ9 · 2023-11-01

**Soundness:** 4 excellent
**Presentation:** 4 excellent
**Contribution:** 3 good
**Rating:** 6
**Confidence:** 4

**Summary:**

The paper systematically studies the effect of the time schedule used for the diffusion model used in score distillation sampling for 3D generation. The key observation is that at different time steps, the denoiser focuses on different types of content -- high-frequency vs. low-frequency. It proposes a schedule to leverage this observation in order to improve the optimization quality and efficiency.

**Strengths:**

I really ike the idea proposed in the paper because it’s simple yet effective in improving the quality of 3D generation from text and images. It studies an aspects of using diffusion model for 3D diffusion model -- how to schedule the timestep used to condition the diffusion model, which is not systematically explored in prior literature. In the paper, the authors show both quantitative and qualitative experiments as well as detailed analysis to demonstrate the effectiveness of this simple technique. The authors also show the generality of this technique by showing results across many state-of-the-art 3D generation methods so it can be easily adopted by the community in any future work along the line.

**Weaknesses:**

1. Though I really like the idea, the paper studies a small aspect of 3D generation which can be seen as a trick to improve results. If more studies can be conducted such as using I to accelerate 2D image generation, the scope of the paper will be further expanded. But I don't think the authors need to conduct these experiments in the rebuttal.
2. The quantitative experiments presented in the paper are limited. While the authors demonstrated the effectiveness of the proposed technique on many different methods with qualitative examples, quantitative evaluation will make these results more convincing.

**Questions:**

The Score Jacobian Chaining paper has studied the scheduling of diffusion timesteps to an extent. Due to its high relevance, I think the paper should discuss the related findings in that paper and put things in context.

---

> ### Author Response · Authors · 2023-11-22
>
> Thank you very much for the insightful suggestions. We’ve provided more results according to your comments.
>
> **W1. Scope of this work.**
>
> Thank you very much for the insightful suggestion. Improving sampling quality and efficiency has been an important aspect for 2D generative models. However, while optimization-based 3D generation is known to be time-consuming with sub-optimal generation results, understanding and addressing the inefficiencies stemming from the formulation with random t-sampling has been overlooked, which is the focus of our work. We totally agree that an unified study on 2D and 3D sampling strategy would be very exciting future work.
>
> **W2. More quantitative experiments.**
>
> We now provide comprehensive user study results in Section E of the appendix.
>
> **Q1. The paper should discuss the findings in the Score Jacobian Chaining paper.**
>
> Thank you very much for the suggestion. SJC did experiment with annealed variance scheduling on 2D image generation, but the observations were inconclusive: annealed t benefits sampling for unconditional distribution (FFHQ and LSUN) but not so much for sampling from stable diffusion. We hypothesize that the difference lies in implementation and time scheduling details. Furthermore, our analysis on both 2D and 3D generation reveals more details of the optimization process for designing an effective time schedule. We'll add more discussions on findings in SJC.

---

### Author Response · Authors · 2023-11-22

We would like to thank all the reviewers for their constructive comments and valuable feedbacks. We are encouraged by the recognition that "I really like the idea proposed because it's simple yet effective", "it can be easily adopted by the community in any future work along the line", "the paper is well-written and presents its ideas clearly", "its experiments are illuminating", "the method is well motivated" and etc.

According to the reviews, we have added sections E-K in appendix (marked as blue in the revised paper).

We individually respond to each reviewer’s comments below. Discussions and results will be added to new revisions.

---

### Meta-Review · Area_Chair_GvGi · 2023-12-07

**Metareview:**

The submission received mixed reviews, with the majority on the positive side. The reviewers appreciate the simplicity of the method with extensive analysis in the experiments. zkfR specifically requested additional experiments and had concerns about the scope, which the AC believes the authors have adequately addressed in the response. After reading the paper, the reviewers' comments, the authors' rebuttal and the discussions, the AC recommends acceptance.

**Justification For Why Not Higher Score:**

Although this paper improves a specific technique in a relatively new field, the scope of the paper is somewhat limited.

**Justification For Why Not Lower Score:**

The paper provides extensive analysis and demonstrates various practical improvements on optimization-based text-to-3D generation, and I believe its merits deserves acceptance.

---

### Decision · Program_Chairs · 2024-01-16

Accept (poster)